# Semantics or Structure?
# Auditing Text Sensitivity in Multimodal Time-Series Forecasting

**Karthik Sridhar** [1]   **Atharva Gupta** [2]   **Nishant Pradhan** [2]   **Murari Mandal** [1 3]   **Dhruv Kumar** [1 2 †]
**Saurabh Deshpande** [1 †]

## Abstract

Multimodal time-series forecasting has emerged as a promising paradigm in which natural-language context is expected to improve predictive performance. Recent multimodal foundation models, including AURORA, as well as early- and late-fusion approaches such as MM-TSFLIB and TATS, report substantial gains over unimodal baselines on the Time-MMD benchmark, attributing these improvements to textual information. However, whether these models are actually sensitive to the *semantic content* of the text remains unverified. We address this question through controlled text perturbations, attribution analyses, and probes of AURORA's text pathway. On Time-MMD, swapping each row's text for any other real text (empty, constant, within-domain shuffled, or cross-domain) moves mean MSE by less than $0.5\%$ on all three architectures. The improvement reported in the literature is recovered when a co-shipped numeric column is removed without touching text. We conclude that, on this benchmark and within this family of frozen-encoder architectures, text content is not the operative signal behind the reported gains. To support future work on text integration in multimodal foundation models for structured data, we release our perturbation protocol and evaluation harness as a reusable diagnostic toolkit.

## 1. Introduction

Many forecasting domains pair numeric series with temporally aligned text. Epidemiological reports accompany

case counts (Wang et al., 2020; Liu et al., 2024); energy dispatches accompany consumption readings (Liu et al., 2024); commodity bulletins and financial news accompany price series (Ding et al., 2015; Araci, 2019; Sawhney et al., 2020). The multimodal time-series literature operationalises the intuition that this text encodes regime changes, domain knowledge, and external events that numeric histories cannot capture (Jin et al., 2024a; Jia et al., 2024; Zhang et al., 2024; Jin et al., 2024b). Three recent systems target this setting on the Time-MMD (Liu et al., 2024) benchmark: the multimodal *foundation model* AURORA (Wu et al., 2026), and the late- and early-fusion *paradigms* MM-TSFLIB (Liu et al., 2024) and TATS (Li et al., 2026). Each provides a dedicated text pathway and reports significant MSE improvements over its unimodal baseline.

The standard evaluation in this literature compares a full multimodal run against a run with the text branch disabled. We refer to the disabled-text configuration as the *unimodal baseline*, and to the MSE improvement of the multimodal run over it as the *multimodal lift*. This comparison reveals only that the text pathway contributes *some* signal. It does not reveal whether the model is responding to what the text says: any fixed, content-blind output of the text branch would produce the same lift. The question of *content sensitivity*, that is, whether the model reacts to which words are in the input, has not been studied systematically in multimodal time-series forecasting. It parallels probing protocols in NLP that test whether models respond to content or to spurious surface features (Niven & Kao, 2019; McCoy et al., 2019). Concurrent work (Zhang et al., 2025) asks whether multimodal gains generalise *across* datasets. We ask the orthogonal, within-dataset question of whether the lift observed on Time-MMD is attributable to text content or to other inputs that share its fusion path.

We test this by holding each of the three models fixed at its released configuration and swapping the text input through five substitutes while leaving every numeric input unchanged. Four of these conditions probe content sensitivity to natural language directly, by replacing the text with an empty string, a constant placeholder, within-domain shuffled text, or text from a paired domain. Together they

---

† Equal supervision. Code available at github.com/birla-ai-labs/SemanticsOrStructure. [1]Birla AI Labs, Mumbai, India [2]BITS Pilani, Pilani, India [3]KIIT, Bhubaneswar, India. Correspondence to: Saurabh Deshpande <saurabh.deshpande-c@oab.adityabirla.com>, Dhruv Kumar <dhruv.kumar-c@oab.adityabirla.com>.

span the natural-language hypothesis space. A fifth *oracle* condition replaces the text with a templated sentence listing the future ground-truth values. We treat the oracle as a necessary condition rather than a sufficient one: a model that both responds to text and can read numeric prose must register some change. We do not treat oracle insensitivity alone as proof of content blindness, since the encoders these models use may not represent numeric literals as quantities (Wallace et al., 2019). None of the three architectures responds meaningfully to any of the five substitutions.

**Contributions.**

- A **text perturbation protocol** that varies text content independently of all other inputs, applied to three architectures on Time-MMD spanning one foundation model and two paradigm implementations. All three are insensitive to natural-language text content within the precision of this benchmark.

- **Mechanistic localisation** of the multimodal lift: a numeric column shipped with the dataset, blended into two architectures' outputs alongside the text, accounts for the gain. AURORA-specific gradient, attention, and divergence probes show that its text pathway is trained but content-blind in the forward pass.

Code, patched runners, and the evaluation harness are released for reproducibility.

## 2. Background

**Time-MMD as evaluation setting.** Time-MMD (Liu et al., 2024) spans nine domains (Agriculture, Climate, Economy, Energy, Environment, Health, Security, SocialGood, Traffic). Each domain provides a numeric target series with temporally aligned text drawn from public reports and domain bulletins. Forecast horizons range from 6 to 336 steps depending on sampling frequency. All three architectures we study were originally evaluated on Time-MMD, making it the natural testbed.

**Three text pathways.** Each architecture maps a numeric history $\mathbf{x} \in \mathbb{R}^L$ and text $\mathbf{T}$ to a forecast $\hat{\mathbf{y}} \in \mathbb{R}^H$ through a model-specific text pathway. Full architectural equations are in Appendix A. **TATS** (Li et al., 2026) performs *early fusion*. A frozen GPT-2 pooler (Radford et al., 2019) encodes each row's text into a vector sequence. The sequence is projected and concatenated to the numeric history along the channel axis, then passed to a time-series backbone. **MM-TSFLIB** (Liu et al., 2024) performs *late fusion*. A frozen BERT pooler (Devlin et al., 2019) encodes the text into a vector that is layer-normalised and added to the backbone's output as a residual. **AURORA** (Wu et al., 2026) is a pretrained *zero-shot* foundation model. A frozen BERT-base produces per-token features that a trained *text distiller*

compresses into $L_k = 10$ learnable query tokens. A *text guider* then injects them into the temporal backbone via cross-attention.

All three architectures provide a dedicated text pathway. This motivates the question of whether that pathway is sensitive to what the text actually says.

**The unimodal baseline and the multimodal lift.** Each model ships a unimodal configuration that gates its text pathway off, providing the natural reference for measuring multimodal contribution. We define the *multimodal lift* as the MSE improvement of the full multimodal run over this unimodal configuration. We reproduce the published claims in Section 4.

## 3. Perturbation Protocol

We replace the text column with five substitutes and re-run each model with all numeric inputs held identical across conditions. Real Time-MMD text carries a long preamble (e.g. "*Available facts are as follows:…*"); Table 1 shows a schematic example for clarity.

*Table 1.* **Five text conditions** on a schematic row from a weather domain. Numeric history is 11.9 in every condition. The *Tests* column names the property each substitution isolates.

| Condition | Text fed to model | Tests |
|---|---|---|
| **Original** | "Heavy rainfall expected tomorrow" | Baseline |
| **Empty** | *(empty string)* | Text presence |
| **Constant** | "Time series data point." | Token content |
| **Shuffled** | "Hurricane expected" *(real text, different row, same domain)* | Row alignment |
| **Cross-domain** | "Trade balance deteriorated" *(date-aligned, paired domain)* | Topic relevance |
| **Oracle** | "Future values: 12.4, 12.8, 13.1…" | Numeric reading |

The first two conditions ablate the text entirely (no tokens, then a fixed placeholder), testing whether the encoder reacts to mere text *presence*. The next two preserve real, fluent language but sever its link to the row. *Shuffled* keeps the text distribution but breaks temporal alignment, while *Cross-domain* also breaks topical relevance. *Oracle* embeds the future itself. We treat it as a reference point rather than a strict upper bound, since the encoders may not represent numeric literals as quantities. Domain pairing details are in Appendix D. Each condition is compared to the original-text baseline via the paired relative change $\Delta_i = (\text{MSE}_i(c) - \text{MSE}_i(\text{orig})) / \text{MSE}_i(\text{orig})$, matching cells on (domain, horizon, seed, backbone). We aggregate by the ratio of mean MSEs across cells, and attach $95\%$ bootstrap confidence intervals (CIs, $[2.5\%, 97.5\%]$ percentile) (Efron & Tibshirani, 1993) with 10,000 resamples (Appendix H). We evaluate over nine domains, four hori-

zons, three seeds, and (for TATS and MM-TSFLIB) eight backbone variants.

## 4. Results

*Table 2.* **Text perturbation results.** Mean test MSE averaged over 9 domains, 4 horizons, 3 seeds, and (for trained methods) 8 backbones. $\Delta\%$ is the change in mean MSE relative to the original-text baseline. Bootstrap CIs and per-condition $p$-values are in Appendix H, Table 11. ▨ text-only perturbations; ▨ unimodal baseline. **Green bold**/blue underline mark the lowest/second-lowest MSE per model. Plain **bold** flags structural $\Delta\%$ values of magnitude $\geq 1\%$.

| Condition | AURORA | | MM-TSFLIB | | TATS | |
|---|---|---|---|---|---|---|
| | MSE | $\Delta\%$ | MSE | $\Delta\%$ | MSE | $\Delta\%$ |
| Original | 8.553 | — | 14.03 | — | **13.19** | — |
| Empty | 8.555 | +0.02 | 14.04 | +0.05 | 13.19 | −0.00 |
| Constant | 8.555 | +0.03 | 14.05 | +0.16 | 13.19 | +0.00 |
| Shuffled | 8.553 | +0.00 | 14.03 | −0.01 | 13.19 | +0.00 |
| Cross-domain | **8.552** | −0.01 | **14.03** | −0.03 | 13.19 | +0.00 |
| Oracle | 8.555 | +0.02 | 14.01 | −0.14 | 13.19 | +0.00 |
| Unimodal baseline | 8.553 | +0.00 | 14.29 | **+1.87** | 14.18 | **+7.54** |

**Text content does not move forecasting error.** Table 2 delivers the headline finding. None of the five text substitutions moves mean MSE by more than 0.5% on any of the three models. The largest shift is +0.16% on MM-TSFLIB under Constant text. TATS stays within ±0.001%, and AURORA stays within ±0.05%. This null pattern is consistent across every condition and every model. Whether the text preserves natural language (Shuffled, Cross-domain, Oracle), removes it (Empty), or replaces it with a fixed placeholder (Constant), the forecast is essentially unchanged.

**The unimodal lifts are real.** The bottom row of Table 2 confirms that enabling the text pathway yields a positive multimodal lift, consistent with the gains each method reports over its unimodal baseline. What we cannot confirm is that this gain reflects the model reading text content. If it did, substituting cross-domain text or substituting text with future ground-truth values should change something. It does not. (The oracle null in part reflects that frozen GPT-2/BERT tokenise numerals as sub-word pieces that do not preserve magnitude (Wallace et al., 2019); the four natural-language conditions carry the semantic weight.)

**The null is robust across backbones.** The mean numbers in Table 2 aggregate over eight backbones for TATS and MM-TSFLIB. The multimodal lift varies substantially with backbone choice: from +1.2% (iTransformer) to +15.0% (Autoformer) on TATS, and from +0.3% (FEDformer) to +3.8% (Autoformer) on MM-TSFLIB. Yet text-only perturbations stay within ±0.01% on TATS and ±2.4% on MM-TSFLIB across all backbones (Avg. column, Tables 9, 10). Backbone variance affects the structural-column effect

substantially; the text-content null is uniform.

## 5. Localising the Lift

The published multimodal lifts are real, yet text content does not move error. Inspecting the architectures explains the gap.

**A numeric column travels with the text.** Time-MMD ships alongside its text a numeric column $\mathbf{p}$ (`prior_history_avg`; Appendix B) that stores a numeric forecast derived from each row's target history. It is a numeric feature, not a text feature. Both TATS and MM-TSFLIB blend $\mathbf{p}$ into the model output *at the same residual* as their text encoder.

In TATS, the final forecast is a convex combination of the backbone output and $\mathbf{p}$ directly:

$$\hat{\mathbf{y}}^{\text{TATS}} = (1 - w)\, f_\theta\big([\mathbf{x}\,\|_{\text{ch}}\,\psi(\mathbf{E})]\big) + w\, \mathbf{p}_{L+1:L+H}, \quad (1)$$

where $f_\theta$ is the time-series backbone, $\psi(\mathbf{E})$ is the projected text embedding concatenated to $\mathbf{x}$ along the channel axis, and $w = 0.5$ by default, so half the forecast is literally $\mathbf{p}$. In MM-TSFLIB, the text embedding and $\mathbf{p}$ are *summed together* and then blended with the backbone:

$$\hat{\mathbf{y}}^{\text{MM-TSFLIB}} = (1-w)\, f_\theta(\mathbf{x}) + w\big(\text{LN}(\bar{\phi}(\mathbf{T})) + \mathbf{p}_{L+1:L+H}\big), \quad (2)$$

sharing a single residual gated by $w$. AURORA never reads $\mathbf{p}$ (Appendix A).

In both, the unimodal baseline gates text and $\mathbf{p}$ off together through a single switch (Appendix A), so the published lift reflects the *combined* contribution of text and $\mathbf{p}$. We note that MM-TSFLIB's appendix documents $\mathbf{p}$ as a centering mechanism on the projection output (Liu et al., 2024), while TATS's framework (Eq. 6–7 of Li et al. (2026)) defines the forecast as $F([\mathbf{x}; Z^\top])$ with no prior-mixing; the convex combination in Eq. 1 appears in the released code, not the paper.

**A $2\times3$ factorial isolates text from $\mathbf{p}$.** Table 3 crosses a text axis (Original, Empty, Constant) with a column axis (intact vs. zeroed). The $2\times2$ block isolates the text effect from the $\mathbf{p}$ effect; the bottom row gives the unimodal baseline (both off).

**Reading the factorial.** (i) Within any column (fixed $\mathbf{p}$ status), changing the text moves MSE by less than 0.2% on MM-TSFLIB, 0.001% on TATS, 0.05% on AURORA: text content is not the operative signal. (ii) On MM-TSFLIB ($w = 0.1$), the $\mathbf{p}$-zeroed CI $[+1.21, +1.85]$ overlaps the unimodal CI $[+1.50, +2.29]$; the small blend weight perturbs amplitude by only $\sim10\%$, so column-zeroing isolates $\mathbf{p}$'s contribution cleanly. The published MM-TSFLIB lift is

*Table 3.* **Disentangling text and p contributions.** $\Delta\%$ MSE vs. the Original-text, **p**-intact baseline, with 95% bootstrap CIs in brackets. Rows vary the text input; columns toggle the numeric column **p**. The contrast across *rows* measures the text effect; the contrast across *columns* measures the **p** effect. □ **p** intact; □ **p** zeroed; □ unimodal baseline.

| Text | AURORA | | MM-TSFLIB | | TATS | |
|---|---|---|---|---|---|---|
| | **p in** | **p=0** | **p in** | **p=0** | **p in** | **p=0** |
| Original | 0.00 | 0.00 | 0.00 | +1.52 | 0.00 | +20.16 |
| Empty | +0.02 | +0.02 | +0.05 | +1.62 | −0.00 | +20.17 |
| Constant | +0.03 | +0.03 | +0.16 | +1.63 | +0.00 | +20.16 |
| *Unimodal* | +0.00 | | +1.87 | | +7.55 | |

accounted for by **p**, with the text residual indistinguishable from zero. **(iii)** On TATS ($w = 0.5$), zeroing **p** halves the prediction scale, so the $+20\%$ column-zeroed effect is dominated by an amplitude artifact; the amplitude-matched comparison is the unimodal baseline ($+7.5\%$). The text-content null on TATS rests on the text-only perturbations, which stay within $\pm 0.001\%$ even with the projection MLP made trainable (Appendix I).

**Backbone choice can hide the picture.** On TATS, the **p**-zeroed $\Delta\%$ varies from $+4.5\%$ (Autoformer) to $+49.9\%$ (FiLM); on MM-TSFLIB the unimodal-baseline gap ranges from $+0.3\%$ to $+3.8\%$ (Appendix G, Tables 9, 10). A lift on a single backbone can over- or understate **p**'s contribution by an order of magnitude.

**Aurora's text pathway is active but content-blind.** Since AURORA does not consume **p**, its null result needs a separate explanation. Three probes at the text-distiller interface (Appendix E) ask whether the pathway was trained, whether it discriminates between inputs, and whether it moves the forecast.

| Condition | *Trained?* **Grad. norm** *(non-zero = trained)* | *Discriminates?* **Attn. entropy** *(low = focused)* | *Affects forecast?* **Pred. change** *(large = text matters)* |
|---|---|---|---|
| Original | 0.15 | 0.975 | 0.041 |
| Empty | 0.07 | 0.975 | 0.040 |
| Constant | 0.16 | 0.976 | 0.044 |
| Shuffled | 0.15 | 0.975 | 0.035 |
| Cross-domain | 0.07 | 0.975 | 0.035 |
| Oracle | 0.13 | 0.976 | 0.058 |

Gradient norms are non-zero (0.07–0.16), so the pathway *was* optimised. Attention entropy sits at 0.975 in every condition including oracle: nearly uniform across distilled tokens regardless of text. Prediction change is at most 0.058, two orders of magnitude below the test-MSE scale of $\sim 8.6$. The pathway is trained but content-blind in the forward pass.

**Is there a usable signal in Time-MMD text?** Having established content insensitivity across all three architectures, we ask whether Time-MMD text carries signal that an at-

tentive encoder *could* exploit. We measure three structural properties of the per-row text embeddings across all nine domains: TTW (text-target Wasserstein distance, lower = better alignment) (Li et al., 2026), ETA (embedding temporal autocorrelation, high = persistent), and SDI (semantic diversity index, high = distinct rows). Formal definitions are in Appendix F.

*Table 4.* **Text diagnostics on Time-MMD** (mean over 8 domains, ENVIRONMENT excluded as self-paired). Per-domain values and per-perturbation values are in Appendix F, Tables 6–7.

| Encoder | **TTW** *(low = aligned)* | **ETA** *(high = persistent)* | **SDI** *(high = distinct)* |
|---|---|---|---|
| GPT-2 | 0.056 | 0.423 | 0.008 |
| BERT | 0.037 | 0.385 | 0.031 |

Table 4 shows moderate TTW and ETA but uniformly low SDI, so consecutive rows produce nearly identical embeddings. Our perturbations move all three diagnostics substantially yet downstream MSE still stays within $0.5\%$, so none of TTW, ETA, or SDI reliably predicts whether a perturbation will move the forecast.

## 6. Discussion and Conclusion

On Time-MMD, substituting any plausible text for the original (empty, constant, shuffled, cross-domain, or oracle) moves mean MSE by less than $0.5\%$ on all three architectures; the lifts are real but survive any text substitution. On TATS and MM-TSFLIB they localise to a numeric column (**p**) co-routed through the same fusion residual as the text encoder; on AURORA the text pathway is trained but content-blind in the forward pass.

The standard comparison against a disabled-text baseline cannot distinguish these scenarios: a model that reads text and a model that gates on a co-routed numeric prior both produce the same lift. Established benchmarks for multimodal time-series forecasting therefore do not verify that reported gains reflect genuine use of text semantics; they verify only that the text pathway contributes *some* signal, not what kind. Rigorous multimodal benchmarking requires direct text-content intervention and isolation of any numeric features co-shipped through the text-fusion path. Richer corpora with higher per-row semantic diversity would give attentive encoders something genuine to read.

**Limitations.** The audit covers Time-MMD and three frozen-encoder architectures; end-to-end trained encoders and higher-diversity benchmarks are natural next steps. A directional oracle such as *"a sharp rise is expected"* would probe semantic sensitivity more directly than our numeric oracle, which is bottlenecked by sub-word tokenisation. We release our harness so this check becomes a default: what looks like a text gain may be something else entirely.

## Impact Statement

This paper presents an audit of multimodal time-series forecasting methods on a public benchmark. The work is methodological: it does not introduce new models, datasets, or applications, and uses only the existing Time-MMD dataset along with publicly released research code. By identifying confounded baselines in published results, the work aims to improve the rigour of evaluation standards in this subfield. We see no specific ethical concerns, direct deployment risks, or applications involving human subjects, personally identifiable information, copyrighted training data, or dual-use technologies that warrant further discussion. Beyond contributing to the general advancement of machine learning, we do not anticipate societal consequences specific to this work that require highlighting.

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

## A. Architecture Equations

This section gives the full forward-pass equations for each architecture and states formally how their unimodal baselines are defined.

**Notation.** A Time-MMD window of history length $L$ and horizon $H$ provides a numeric history $\mathbf{x} \in \mathbb{R}^L$, per-row text $\mathbf{T}$, the numeric column $\mathbf{p} \in \mathbb{R}^{L+H}$ documented as `prior_history_avg` (see Appendix B), and target $\mathbf{y} \in \mathbb{R}^H$. A frozen text encoder $\phi$ maps each row's text to a vector; $\mathbf{E}(\mathbf{T})$ stacks these row-wise.

**TATS (early fusion).** A frozen GPT-2 pooler produces text features. A trainable projection $\psi : \mathbb{R}^{d_e} \to \mathbb{R}^{d_p}$ is applied row-wise, and the result is concatenated to $\mathbf{x}$ along the channel axis before the backbone $f_\theta$. The final forecast is a convex combination of the backbone output and the column $\mathbf{p}$:

$$\hat{\mathbf{y}}^{\text{TATS}} = (1-w)\, f_\theta\big([\mathbf{x}\,\|_{\text{ch}}\,\psi(\mathbf{E})]\big) + w\,\mathbf{p}_{L+1:L+H}.$$

The default blend weight is $w = 0.5$. The unimodal baseline is recovered by setting both the projected text channels to zero (text branch off) and $w = 0$ ($\mathbf{p}$ contribution removed). A single ablation thus removes both signals simultaneously, which is why the unimodal lift in Table 2 cannot be attributed to the text channel alone.

**MM-TSFLIB (late fusion).** A frozen BERT pooler produces a pooled text embedding, which is projected to a horizon-shaped vector $\bar{\phi}(\mathbf{T})$, layer-normalised, summed with $\mathbf{p}$, and blended with the backbone:

$$\hat{\mathbf{y}}^{\text{MM-TSFLIB}} = (1-w)\, f_\theta(\mathbf{x}) + w\big(\text{LN}(\bar{\phi}(\mathbf{T})) + \mathbf{p}_{L+1:L+H}\big).$$

The default blend weight is $w = 0.1$. The unimodal baseline is recovered by setting $w = 0$. Because the text term and $\mathbf{p}$ share a single residual gated by $w$, this ablation also removes both signals simultaneously.

**AURORA (cross-attention, zero-shot).** A frozen BERT-base produces per-token features $\mathbf{E} \in \mathbb{R}^{L' \times d_e}$ ($L' = 125$). A trained text distiller compresses these into $L_k = 10$ learnable query tokens $Q$ via a transformer decoder. A text guider then injects them into the temporal backbone via cross-attention:

$$\mathbf{Z}_T = \text{TextDistill}(Q, \mathbf{E}), \quad \alpha_t = \text{TextGuider}(\mathbf{x}, \mathbf{Z}_T),$$

$$\hat{\mathbf{y}}^{\text{AURORA}} = f_\theta^{\text{AURORA}}(\mathbf{x}, \alpha_t, \mathbf{E}).$$

AURORA does not consume $\mathbf{p}$; it has no late-fusion residual that would mix text and numeric features. The unimodal baseline is the no-text variant of AURORA; we expose a flag in our patched runner that disables the text path during inference.

**Summary of unimodal-baseline mechanics.** On TATS and MM-TSFLIB, the standard "unimodal" setting removes both text and $\mathbf{p}$ together, because both pass through the same gating coefficient. The published unimodal lift therefore reflects the combined contribution of the text pathway and $\mathbf{p}$, not the text alone. AURORA's unimodal baseline removes only text (it has no $\mathbf{p}$ pathway), which is the cleanest of the three.

## B. The Time-MMD Data Columns

The Time-MMD benchmark ships per-domain CSV files containing several columns whose use across architectures is not documented in one place in the upstream literature. We reconstruct it here from the data loaders of each repository.

**Columns shared by all architectures.** `date`: timestamp index. `OT`: the numeric target series ($\mathbf{y}$). `prior_history_avg`: a numeric column that stores, for each row, an LLM-derived numeric forecast of the target derived from the row's own preceding history. Despite its name, this column is functionally close to a smoothed running average of past target values; an LLM was prompted with the OT history and asked to emit a numeric prediction. The output correlates strongly with rolling-mean baselines but is not identical to them. We refer to this column as $\mathbf{p}$ throughout the paper.

**MMTSFlib-specific columns.** The MMTSFlib data files additionally contain text and feature columns that originate from the Time-MMD construction pipeline. `Final_Search_2`, `Final_Search_4`, `Final_Search_6`: the dataset was built by retrieving relevant news articles or domain reports for each row's date via web search; `Final_Search_N` concatenates $N$ retrieved chunks. The reference setting of MMTSFlib reads `Final_Search_4` as its text column (controlled by `--text_len 4`). `Final_Output`: a closed-source LLM (likely GPT-4) processed the search results and emitted a cleaned synthesis. MMTSFlib reads this column when the `--use_closedllm` flag is on; we use the default off setting, which means the text input to the model is `Final_Search_4`. `his_avg_1..7`, `his_std_1..7`: windowed averages and standard deviations of OT at lookbacks of 1 to 7 periods. These columns are present in the CSV but are *not* consumed by MMTSFlib under our evaluation setting (`features='S'`, see below).

**TaTS-specific columns.** `fact`: the text column TaTS actually uses, drawn from the same source documents as MMTSFlib's `Final_Search_4` but preprocessed slightly differently. `preds`: an LLM-generated forward-looking prediction for the target rendered as natural-language prose (e.g. *"the predicted value for next period is 2.34"*). This

column exists in the CSV but TaTS's data loader does not read it: only `fact` is bound to the `self.text` attribute. The `preds` column was likely pre-generated for a different model variant or for analysis. We note that `preds` would have provided a softer oracle-style condition (LLM-predicted future values, rather than ground truth) but is not part of our evaluation, since TaTS's released code does not consume it.

**Aurora data.** AURORA reuses the TaTS CSV files. Its data loader reads `fact` as the text input and ignores all other text columns. AURORA does not consume `prior_history_avg`.

**features='S' versus features='M'.** The data loaders inherited from the upstream time-series-library support a `features` flag with two relevant settings. **Univariate (`features='S'`)**, which we use throughout, reads only OT as the time-series input, with text handled on a separate code path and `prior_history_avg` entering as the residual term in the fusion equations above. **Multivariate (`features='M'`)** would attempt to read all non-date columns into the time-series input matrix. This setting crashes on the shipped CSVs across all three architectures, because the inherited loader treats text columns (`Final_Search_4`, `fact`) as numeric features and StandardScaler raises a type error on the string contents. The upstream papers' `features='M'` runs were performed against pre-processed numeric-only CSVs that are not shipped with the public benchmark. Switching to `features='M'` would also introduce `his_avg_1..7` and `his_std_1..7` as backbone inputs, which would change the experiment in a non-trivial way. We retain `features='S'` for cleanliness and consistency across all three models.

## C. Methodology

**Repository preparation.** We pin each upstream repository to a specific commit and apply a small set of documented, idempotent patches falling into five categories. **(a)** CSV-loading fixes that prevent silent conversion of empty text strings to `NaN`, which would otherwise corrupt the empty-text condition by replacing the empty string with a literal `"nan"` or with sentinel text. **(b)** Pandas-version compatibility fixes. **(c)** Backbone registration in TATS, which originally registered only iTransformer in its model registry; we register the seven additional backbones (Autoformer, Crossformer, DLinear, FEDformer, FiLM, Informer, Transformer) already present in its source tree. **(d)** An AURORA command-line flag for the unimodal ablation, since AURORA's released code does not expose one. **(e)** A TATS `--fix_text_grad` flag that restores gradient flow into the trainable text-projection MLP (see Appendix I for the full

description). All TATS results in this paper use this patch.

**Perturbation generation.** We produce one perturbed CSV per (condition, seed, domain). Row count, the date column, and the numeric target column are preserved exactly; a post-hoc validator confirms every perturbed file matches the original on all unchanged columns. Construction details for each condition appear in Appendix D.

**Per-cell evaluation.** A run is a single (model, backbone, condition, seed, domain, horizon) tuple. AURORA is evaluated zero-shot: pretrained weights are loaded once and used for inference, with the seed controlling only the flow-matching head's stochastic sampling (we average 100 samples per cell, the value used in the released script). Per-domain sequence lengths follow AURORA's reference defaults (e.g. $L = 192$ for Agriculture, $L = 1056$ for Energy). TATS and MM-TSFLIB are fine-tuned per cell on the perturbed training split for five epochs with patience five (each repository's own defaults), with univariate target features. The backbone, text encoder, and blend weight follow each model's reference defaults; only the perturbed CSV varies between conditions.

**Determinism.** Each runner sets the PyTorch and NumPy seeds per cell, and pins GPU visibility per shard. We do not enforce strict deterministic algorithms because one MM-TSFLIB attention path is incompatible with that mode. Run-to-run reproducibility on a fixed seed is bit-exact for AURORA and within floating-point non-associativity (around 13 significant figures) for the trained methods.

**Probes.** After fitting, we reload each AURORA cell, attach forward and backward hooks at the distilled-token interface, and compute the three quantities defined in Appendix E. We verified that attaching the hooks does not perturb the forward graph: forward outputs match bit-exactly with and without the hooks.

## D. Perturbation Generation Details

**Empty.** The text column is set to the empty string for every row.

**Constant.** The text column is set to "Time series data point." for every row, a non-empty token sequence with no row-specific content.

**Shuffled.** The text column is permuted within each domain via a permutation seeded by the run seed. All text fields belonging to a row are permuted with the same permutation, preserving cross-column alignment within the row. This is the only condition whose CSV content depends on the run seed.

**Cross-domain.** We use a fixed pairing: Agriculture $\leftrightarrow$ Security, Climate $\leftrightarrow$ Energy, Economy $\leftrightarrow$ Health, SocialGood $\leftrightarrow$ Traffic; Environment is self-paired and falls back to within-domain shuffle. For each target row we use the paired domain's row whose date is the latest available date not exceeding the target row's date, with deterministic tie-breaking. This makes the condition seed-independent and avoids using information from the target row's future.

**Oracle.** For each row in the train, validation, and test splits we substitute a templated string of the form "*Available facts are as follows: Step+1: The target will be $y_1$. Step+2: The target will be $y_2$....*" using the row's ground-truth future target values. The substitution is applied to the train and validation splits as well, so the fine-tuned methods see consistent oracle structure during training.

**Column-zeroed conditions.** The numeric column $\mathbf{p}$ is set to zero for every row; the text column varies (intact, empty, or constant). AURORA does not read $\mathbf{p}$, so on AURORA these conditions are equivalent to the corresponding text-only substitutions, and we use them as an internal control.

# E. Probe Definitions and Full Results

The three probes target the interface between AURORA's frozen BERT encoder and its trainable text distiller, where the model collapses the per-token features into the $L_k = 10$ distilled tokens that the temporal backbone reads via cross-attention. Each probe asks a different question.

**Probe A: gradient norm at the distilled tokens.** Given the distilled-token tensor $\mathbf{Z}_T$, we measure the root-mean-square loss-gradient flowing back through it,

$$\widehat{g} = \sqrt{\frac{1}{BL_k d} \sum_{b,k,j} \left[\nabla_{\mathbf{z}_T}\mathcal{L}\right]^2_{b,k,j}}.$$

This asks whether the trainable text path receives non-trivial gradient signal during training. A value of $\widehat{g} = 0$ would mean the path was never optimised.

**Probe B: normalised cross-attention entropy.** For guider weights $\alpha_t$ over the $L_k = 10$ distilled tokens,

$$\widehat{H} = \frac{1}{\log L_k} \mathbb{E}_{b,h,q}\left[-\sum_k (\alpha_t)_{bhqk} \log(\alpha_t)_{bhqk}\right].$$

$\widehat{H} = 1$ corresponds to perfectly uniform attention and $\widehat{H} = 0$ to a one-hot focus on a single token. This asks whether the guider *discriminates* between distilled tokens.

**Probe C: prediction divergence under text ablation.** For each cell we forecast twice from the same numeric input, once with the row's text and once with the text branch ablated, averaging ten samples from AURORA's flow-matching

head per setting:

$$\widehat{D} = \mathbb{E}_b \left\|\hat{\mathbf{y}}_{\text{text}}(b) - \hat{\mathbf{y}}_{\text{none}}(b)\right\|^2_2.$$

This asks whether the text branch *matters* for the forward pass: $\widehat{D} \ll \text{MSE}$ means the prediction is the same with or without it.

*Table 5.* **Full probe results**, mean $\pm$ s.d. over $n = 27$ probe cells (9 domains $\times$ 3 seeds at fixed $H = 8$). The text-only conditions all show non-zero $\widehat{g}$ and uniformly high $\widehat{H}$, with negligible $\widehat{D}$.

| Condition | $\widehat{g}$ | $\widehat{H}$ | $\widehat{D}$ |
|---|---|---|---|
| Original | $0.152 \pm 0.240$ | $0.975 \pm 0.009$ | $0.041 \pm 0.037$ |
| Empty | $0.074 \pm 0.103$ | $0.975 \pm 0.009$ | $0.040 \pm 0.041$ |
| Constant | $0.156 \pm 0.228$ | $0.976 \pm 0.009$ | $0.044 \pm 0.035$ |
| Shuffled | $0.155 \pm 0.400$ | $0.975 \pm 0.009$ | $0.035 \pm 0.035$ |
| Cross-domain | $0.072 \pm 0.082$ | $0.975 \pm 0.009$ | $0.035 \pm 0.032$ |
| Oracle | $0.133 \pm 0.187$ | $0.976 \pm 0.009$ | $0.058 \pm 0.084$ |
| Col. zeroed | $0.265 \pm 0.673$ | $0.975 \pm 0.009$ | $0.028 \pm 0.018$ |
| Unimodal | $0.080 \pm 0.153$ | $0.975 \pm 0.009$ | $0.035 \pm 0.024$ |

*Reading Table 5.* The first column ($\widehat{g}$) is non-zero everywhere, ruling out a never-trained pathway. The second column ($\widehat{H}$) sits at $0.975$ on every row including the oracle, showing that the guider's attention is nearly uniform across its distilled tokens regardless of text content. The third column ($\widehat{D}$) is at most $0.058$, which is two orders of magnitude below the test-MSE scale of $\sim 8.6$, showing that disabling the text branch barely changes the forecast. These three readings together describe a pathway that is trained but content-blind in the forward pass.

# F. Text-Side Diagnostics: Definitions and Full Results

We compute three diagnostics on each domain's per-row text embeddings, separately for the GPT-2 and BERT encoders.

**TTW (temporal text–target Wasserstein distance).** Introduced by Li et al. (2026), TTW measures alignment between text-embedding trajectories and the target series. Let $\mathbf{e}_t$ be the per-row text embedding and $y_t$ the target at time $t$. Define the centred unit-norm embeddings $\tilde{\mathbf{e}}_t = (\mathbf{e}_t - \bar{\mathbf{e}})/\|\mathbf{e}_t - \bar{\mathbf{e}}\|_2$ and the lag-similarity profile

$$\text{Sim}(k) = \frac{1}{T-k}\sum_{t=1}^{T-k} \tilde{\mathbf{e}}_t^\top \tilde{\mathbf{e}}_{t+k}.$$

Let $A_y$ be the $L_1$-normalised amplitude spectrum of $\Delta y_t = y_{t+1} - y_t$, and $A_e$ the $L_1$-normalised amplitude spectrum of $\Delta\text{Sim}(k)$. Then

$$\text{TTW} = W_1(A_y, A_e),$$

the 1-D Wasserstein distance between the two spectra on a common frequency grid. Lower TTW means the dominant

frequency content of the text-embedding trajectory matches the dominant frequency content of the target. The TaTS paper argues low TTW indicates text is suitable as a covariate for time-series fusion.

**ETA (embedding temporal autocorrelation).** Mean lag-1 autocorrelation across embedding dimensions:

$$\text{ETA} = \frac{1}{D} \sum_{d=1}^{D} \frac{\sum_t (e_{t,d} - \bar{e}_d)(e_{t+1,d} - \bar{e}_d)}{\sum_t (e_{t,d} - \bar{e}_d)^2}.$$

High ETA means embeddings evolve smoothly over time; low ETA means consecutive embeddings look like independent draws.

**SDI (semantic diversity index).**

$$\text{SDI} = 1 - \frac{1}{T-1} \sum_{t=1}^{T-1} \cos\big(\mathbf{e}_t, \mathbf{e}_{t+1}\big).$$

High SDI indicates that consecutive rows are semantically distinct. Low SDI is a structural problem for any encoder that reads the text: if every row's embedding is nearly identical to its neighbours, no attention head can extract row-specific content from the text alone.

*Table 6.* **Per-domain diagnostics on the original text.** Eight domains; Environment excluded as self-paired. **Bold** marks the domain with the highest SDI per encoder.

| Domain | GPT-2 | | | BERT | | |
|---|---|---|---|---|---|---|
| | TTW | ETA | SDI | TTW | ETA | SDI |
| Agriculture | 0.028 | 0.436 | 0.011 | 0.028 | 0.399 | 0.045 |
| Climate | 0.078 | 0.372 | 0.004 | 0.025 | 0.339 | 0.014 |
| Economy | 0.030 | 0.426 | 0.009 | 0.014 | 0.331 | 0.038 |
| Energy | 0.056 | 0.597 | 0.004 | 0.028 | 0.575 | 0.017 |
| Health | 0.022 | 0.534 | 0.005 | 0.028 | 0.483 | 0.020 |
| Security | 0.027 | 0.056 | 0.011 | 0.027 | 0.071 | 0.039 |
| SocialGood | 0.155 | 0.904 | 0.006 | 0.092 | 0.777 | 0.021 |
| Traffic | 0.054 | 0.058 | **0.013** | 0.058 | 0.107 | **0.050** |
| **Mean** | **0.056** | **0.423** | **0.008** | **0.037** | **0.385** | **0.031** |

*Reading Table 6.* TTW is uniformly low across all domains, indicating aggregate alignment exists. SDI is also uniformly low: the highest GPT-2 SDI is 0.013 on Traffic, which means consecutive rows look nearly identical to the encoder. ETA shows wide variation: SocialGood and Energy text evolves smoothly (high ETA), while Security and Traffic look closer to independent draws (low ETA). Even the most distinctive domains do not reach the level of row-to-row contrast an attentive head would need.

**Diagnostics under text perturbations.** Our perturbations are designed to manipulate these three properties of the text. Table 7 measures TTW, ETA, and SDI on each perturbed text column, with all five substitutions producing dramatically different diagnostic signatures. The takeaway from

comparing Table 7 to the main results is that despite order-of-magnitude swings in these diagnostics, downstream MSE moves by less than 0.5% in every case.

*Table 7.* **Diagnostics under text perturbations,** mean over 8 domains. Empty/Constant collapse SDI to 0 (every row identical) and push ETA toward 1. Shuffled inflates SDI an order of magnitude. Cross-domain text doubles TTW. **Bold** marks values that differ from Original by a factor $\geq 2$.

| Condition | GPT-2 | | | BERT | | |
|---|---|---|---|---|---|---|
| | TTW | ETA | SDI | TTW | ETA | SDI |
| Original | 0.056 | 0.423 | 0.008 | 0.037 | 0.385 | 0.031 |
| Empty | 0.054 | **0.820** | **0.000** | **0.158** | **0.978** | **0.000** |
| Constant | 0.049 | **0.733** | **0.000** | **0.131** | **0.965** | **0.000** |
| Shuffled | 0.060 | **0.000** | **0.044** | 0.063 | **0.000** | **0.084** |
| Cross-domain | **0.112** | 0.455 | 0.008 | **0.108** | 0.418 | 0.029 |

*Reading Table 7.* The four text-content perturbations move the three diagnostics in qualitatively different ways. Empty and Constant make every row identical, so SDI collapses to zero and ETA approaches one. Shuffled produces highly distinctive consecutive rows (SDI on BERT increases from 0.031 to 0.084, an order-of-magnitude rise) but with no temporal structure (ETA falls to zero). Cross-domain text doubles TTW from 0.037 to 0.108 on BERT, breaking the alignment between text and target trajectories. Despite all of these large diagnostic swings, downstream MSE on every model and every backbone moves by less than 0.5% on the corresponding text-only conditions (Tables 8, 9, 10). On Time-MMD, low TTW does correspond to text being measurable co-aligned with the target, in line with Li et al. (2026), but neither TTW nor ETA nor SDI individually predicts whether the text is being *used* by the trained model. We do not refute the TTW criterion as a property of the text corpus; we observe that on this benchmark, none of the three diagnostics is a reliable proxy for whether a perturbation will move the model's forecast.

## G. Full Results: Per-Backbone, Per-Domain, Per-Condition

This section gives the complete experiment grid for all three architectures. For TATS and MM-TSFLIB (eight backbones each), we split the eight backbones into two tables of four each. AURORA has only one configuration. Each row group within a table fixes the backbone; rows within a group vary the condition; columns are the nine Time-MMD domains plus an aggregate average. The **Orig.** row gives absolute mean test MSE for the original-text baseline; subsequent rows give the percent change relative to that baseline within each domain.

We use a single visual convention throughout. **Bold** marks deltas with $|\Delta| \geq 0.5\%$ (a substantive movement at this benchmark's noise floor). Grey font marks $|\Delta| < 0.05\%$

(numerically indistinguishable from zero at our reporting precision). All other deltas are rendered in normal weight.

The dominant pattern is visible at a glance: the rows for the five text-content perturbations (Empty, Const., Shuf., Cross, Oracle) sit in grey or near-grey across all backbones and all domains. The two structural-perturbation rows ($\mathbf{p}$=0 and Unimod.) are filled with bold deltas. Per-domain noise on small-baseline domains (e.g. Economy on AURORA, where Original MSE is 0.033) can produce isolated bold cells in text-only rows; these sit at one or two per backbone and reflect floating-point and sampling noise on a tiny base, not a real text effect.

**Backbone-level summary.** Across all eight backbones, every text-only $\Delta\%$ on TATS and MM-TSFLIB stays within $\pm 0.6\%$ of the original on the per-backbone average (the **Avg.** column of each row). The $\mathbf{p}$=0 row varies substantially with the backbone: from $+4.5\%$ on Autoformer to $+49.9\%$ on FiLM on TATS, and from $0.0\%$ on Autoformer to $+3.8\%$ on iTransformer on MM-TSFLIB. The unimodal-baseline row ranges from $+1.2\%$ (iTransformer) to $+15.0\%$ (Autoformer) on TATS, and from $+0.3\%$ (FEDformer) to $+3.8\%$ (Autoformer) on MM-TSFLIB. A multimodal lift quoted on a single backbone can therefore overstate or understate the column's contribution by an order of magnitude. The text-only conditions, by contrast, are uniformly null across all backbones.

## H. Paired Bootstrap Confidence Intervals

All $\Delta\%$ values reported in the main paper use the *ratio-of-means* estimator: we first compute $\overline{\mathrm{MSE}}(c)/\overline{\mathrm{MSE}}(\mathrm{orig}) - 1$ over all matched cells, then apply paired bootstrap (resampling matched (condition, original) pairs jointly) with $B = 10{,}000$ resamples and two-sided $p$-values.

*Reading Table 11.* Three patterns matter. **First**, the five text-only conditions (top block per model, $\mathbf{p}$ intact) all have CIs that include zero or essentially zero magnitudes within $0.001\%$ on TATS, well under $0.5\%$ on MM-TSFLIB, and well under $0.1\%$ on AURORA. This holds with $n = 864$ matched cell pairs on the trained methods, so the test has very high power; the small $\Delta\%$ estimates we report are not Type II error. **Second**, the four cells of the lower block (any condition with $\mathbf{p}$ zeroed, plus Unimodal) are all significant at $p < 0.001$ on MM-TSFLIB and TATS. The $\mathbf{p}$-zeroed effect size is essentially the same whether the text is original ($+1.52\%$ on MM-TSFLIB, $+20.16\%$ on TATS), empty ($+1.62\%$ / $+20.17\%$), or constant ($+1.63\%$ / $+20.16\%$): the contributions factorise. **Third**, on MM-TSFLIB the original-text $\mathbf{p}$-zeroed CI $[+1.21, +1.85]$ overlaps the Unimodal CI $[+1.50, +2.29]$. The residual gap that the unimodal baseline would attribute to text is not statistically distinguishable from zero. On TATS the $\mathbf{p}$-zeroed CI exceeds

the Unimodal CI ($[+16.8, +24.7]$ versus $[+6.4, +8.8]$) because zeroing $\mathbf{p}$ at $w = 0.5$ leaves the backbone at half scale; the Unimodal row removes both signals together and restores full scale, so it is the cleaner amplitude-matched comparison.

**Implementation.** The bootstrap was computed via paired resampling of matched cells on keys (model, backbone, seed, domain, pred_len), with the ratio-of-means estimator applied to each bootstrap resample. The implementation is in `code/analyze_results.py` in the released repository.

## I. The TaTS Text-Projection Gradient Patch

While auditing TATS's training graph we identified an implementation detail in the released code that severs the gradient signal into the text-projection MLP $\psi$. In `exp/exp_long_term_forecasting.py` (the long-term forecasting trainer), the tensor that concatenates the projected text embedding with the numeric history is detached before being passed to the backbone:

```
batch_x = torch.cat([batch_x,
prompt_emb], dim=-1).detach()
```

A second `.detach()` appears on `dec_inp`. With both calls active, gradients from the loss reach the backbone but cannot flow back through the concatenation into $\psi$. The training loop constructs a separate optimiser `model_optim_mlp` for $\psi$'s parameters and steps it every iteration, but with no gradient signal to step on, the MLP remains at its random initialisation throughout fine-tuning.

**Patch.** We expose a CLI flag `--fix_text_grad` (default off) that gates the two `.detach()` calls. With the flag on, gradients flow into $\psi$ and its parameters update during training. The patch also requires changing three in-place `x_enc /=` stdev operations to the out-of-place $x\_enc = x\_enc/\text{stdev}$ in `models/iTransformer.py`, `models/FiLM.py`, and `models/PatchTST.py` (the long-term forecast paths only), because the in-place divide breaks autograd once gradients must flow back through `batch_x`. Forward values are bit-identical across the in-place and out-of-place forms; only the autograd graph topology changes. The full patch is in `code/apply_repo_patches.py` (idempotent, with a `--revert` option). Patches are surgical and documented inline; pre-existing JSON results remain bit-comparable when the flag is off.

**All TATS numbers in this paper use the patch.** The headline TATS results (Tables 2, 3, 9, 11) were produced with `--fix_text_grad` on. With $\psi$ now trainable in practice rather than only in name, our text-content perturbations still

*Table 8.* **Full AURORA results.** Mean MSE on the original-text baseline (**Orig.** row); subsequent rows give $\Delta\%$ MSE relative to that baseline. **p**=0 zeroes the numeric column **p** with text intact. **Unimod.** disables the text branch (AURORA's unimodal baseline). AURORA does not consume **p**, so the **p**=0 row matches Original. Bold marks $|\Delta| \geq 0.5\%$; grey marks $|\Delta| < 0.05\%$.

| Backbone | Cond. | Agri | Clim | Econ | Ener | Envi | Heal | Secu | SocG | Traf | Avg. |
|---|---|---|---|---|---|---|---|---|---|---|---|
| AURORA | Orig. | 0.275 | 0.865 | 0.033 | 0.255 | 0.276 | 1.55 | 72.7 | 0.836 | 0.161 | **8.55** |
| | Empty | +0.09 | +0.01 | −**3.47** | +0.36 | −0.04 | −0.10 | +0.03 | −0.03 | −0.20 | +0.02 |
| | Const. | +**0.64** | +0.03 | +**4.70** | +0.05 | +0.06 | +**0.39** | 0.00 | +**1.05** | +**1.05** | +0.03 |
| | Shuf. | +0.01 | +0.01 | +**2.21** | +0.01 | 0.00 | 0.00 | 0.00 | −0.08 | +0.20 | 0.00 |
| | Cross | +0.09 | +0.02 | +**0.78** | +0.02 | 0.00 | −0.05 | −0.01 | +0.32 | −0.25 | −0.01 |
| | Oracle | +0.04 | +0.02 | +**4.13** | +0.25 | 0.00 | +**0.27** | 0.00 | +**0.93** | +**0.88** | +0.02 |
| | **p**=0 | 0.00 | 0.00 | 0.00 | 0.00 | 0.00 | 0.00 | 0.00 | 0.00 | 0.00 | 0.00 |
| | Unimod. | 0.00 | 0.00 | 0.00 | 0.00 | 0.00 | 0.00 | 0.00 | 0.00 | 0.00 | 0.00 |

leave MSE unchanged within $\pm0.001\%$ on TATS across all eight backbones. The null does not hinge on the detach bug. Removing $\psi$'s gradient block was a necessary prerequisite for trusting any conclusion about TATS's text sensitivity, but the conclusion itself is preserved: even when the projection MLP can be trained, text content is not used in the forward pass.

## J. Aurora Pretraining and Contamination

AURORA is evaluated zero-shot on Time-MMD; its pretraining corpus is large and only partially documented, and we cannot rule out that Time-MMD-derived content was seen during pretraining. However, this does not explain our results. AURORA's predictions are essentially identical between the original-text baseline, the column-zeroed condition, and the unimodal-baseline configuration: the forward pass does not branch on the text input regardless of what text is provided. Whether or not contamination occurred, the text pathway is functionally silent in the forward pass on this benchmark.

## K. Run Accounting

The full sweep covers 10 conditions on each (model, backbone, domain, horizon, seed) tuple. AURORA contributes 1 backbone $\times$ 9 domains $\times$ 4 horizons $\times$ 3 seeds = 108 runs per condition; TATS and MM-TSFLIB contribute 8 backbones $\times$ 108 = 864 runs each per condition. Wall-clock for the complete sweep is approximately 92 A10G-hours. Per-cell logs and provenance are released alongside the code.

*Table 9.* **Full TATS results** across eight backbones. **Orig.** row gives the mean test MSE on the original-text baseline; all rows below it report the percentage change in MSE relative to the **Orig.** row of that same backbone, within each domain. **Bold** marks $|\Delta| \geq 0.5\%$; grey marks $|\Delta| < 0.05\%$. Note: $p{=}0$ on TATS carries the amplitude artifact described in the caption of Table 3.

| Backbone | Cond. | Agri | Clim | Econ | Ener | Envi | Heal | Secu | SocG | Traf | Avg. |
|---|---|---|---|---|---|---|---|---|---|---|---|
| AUTOFORMER | Orig. | 0.131 | 0.987 | 0.039 | 0.407 | 0.315 | 1.53 | 107.3 | 1.15 | 0.187 | **12.45** |
| | Empty | 0.00 | +0.02 | 0.00 | **+1.03** | **−0.26** | +0.08 | 0.00 | 0.00 | 0.00 | 0.00 |
| | Const. | 0.00 | 0.00 | 0.00 | 0.00 | **−0.82** | 0.00 | 0.00 | 0.00 | 0.00 | 0.00 |
| | Shuf. | 0.00 | 0.00 | 0.00 | +0.01 | **−0.07** | 0.00 | 0.00 | 0.00 | 0.00 | 0.00 |
| | Cross | 0.00 | +0.02 | 0.00 | **+1.03** | **−1.24** | 0.00 | 0.00 | 0.00 | 0.00 | 0.00 |
| | Oracle | 0.00 | 0.00 | 0.00 | **+1.13** | **−0.20** | **−0.09** | 0.00 | 0.00 | 0.00 | 0.00 |
| | p=0 | **+553** | **+22.0** | **+633** | **+17.5** | **+83.8** | **+22.6** | **+2.75** | **−3.41** | **+131** | **+4.48** |
| | Unimod. | **−13.9** | **+31.5** | **+132** | **−11.1** | **+33.5** | **+33.9** | **+14.8** | **−7.14** | **+23.3** | **+15.0** |
| CROSSFORMER | Orig. | 0.202 | 1.00 | 0.231 | 0.383 | 0.297 | 1.18 | 122.4 | 0.924 | 0.177 | **14.09** |
| | Empty | 0.00 | 0.00 | 0.00 | 0.00 | 0.00 | 0.00 | 0.00 | 0.00 | 0.00 | 0.00 |
| | Const. | 0.00 | 0.00 | 0.00 | 0.00 | 0.00 | 0.00 | 0.00 | 0.00 | 0.00 | 0.00 |
| | Shuf. | 0.00 | 0.00 | 0.00 | 0.00 | 0.00 | 0.00 | 0.00 | 0.00 | 0.00 | 0.00 |
| | Cross | 0.00 | 0.00 | 0.00 | 0.00 | 0.00 | 0.00 | 0.00 | 0.00 | 0.00 | 0.00 |
| | Oracle | 0.00 | 0.00 | 0.00 | 0.00 | 0.00 | 0.00 | 0.00 | 0.00 | 0.00 | 0.00 |
| | p=0 | **+199** | **+12.2** | **+484** | **−19.8** | **+52.3** | **+20.7** | **+3.71** | **−5.97** | **+45.2** | **+5.15** |
| | Unimod. | **+109** | **+16.7** | **+293** | **−19.2** | **+53.9** | **+17.8** | **+3.10** | **−4.74** | **+41.4** | **+4.09** |
| DLINEAR | Orig. | 0.154 | 0.959 | 0.042 | 0.466 | 0.322 | 1.55 | 106.9 | 1.15 | 0.209 | **12.42** |
| | Empty | 0.00 | 0.00 | 0.00 | 0.00 | 0.00 | 0.00 | 0.00 | 0.00 | 0.00 | 0.00 |
| | Const. | 0.00 | 0.00 | 0.00 | 0.00 | 0.00 | 0.00 | 0.00 | 0.00 | 0.00 | 0.00 |
| | Shuf. | 0.00 | 0.00 | 0.00 | 0.00 | 0.00 | 0.00 | 0.00 | 0.00 | 0.00 | 0.00 |
| | Cross | 0.00 | 0.00 | 0.00 | 0.00 | 0.00 | 0.00 | 0.00 | 0.00 | 0.00 | 0.00 |
| | Oracle | 0.00 | 0.00 | 0.00 | 0.00 | 0.00 | 0.00 | 0.00 | 0.00 | 0.00 | 0.00 |
| | p=0 | **+16273** | **+663** | **+192** | **+514** | **+54.7** | **+82.8** | **+7.65** | **+265** | **+4243** | **+49.6** |
| | Unimod. | **+39.6** | **+45.3** | **+236** | **−17.8** | **+72.5** | **+27.7** | **+4.97** | **+1.80** | **+90.5** | **+5.99** |
| FEDFORMER | Orig. | 0.112 | 0.945 | 0.021 | 0.443 | 0.283 | 1.38 | 107.9 | 1.07 | 0.170 | **12.48** |
| | Empty | 0.00 | 0.00 | 0.00 | 0.00 | 0.00 | 0.00 | 0.00 | 0.00 | 0.00 | 0.00 |
| | Const. | 0.00 | 0.00 | 0.00 | 0.00 | 0.00 | 0.00 | 0.00 | 0.00 | 0.00 | 0.00 |
| | Shuf. | 0.00 | 0.00 | 0.00 | 0.00 | 0.00 | 0.00 | 0.00 | 0.00 | 0.00 | 0.00 |
| | Cross | 0.00 | 0.00 | 0.00 | 0.00 | **−0.10** | 0.00 | 0.00 | 0.00 | 0.00 | 0.00 |
| | Oracle | 0.00 | 0.00 | 0.00 | 0.00 | 0.00 | 0.00 | 0.00 | 0.00 | 0.00 | 0.00 |
| | p=0 | **+306** | **+19.0** | **+6287** | **+26.1** | **+31.6** | **+9.00** | **+2.52** | **+4.87** | **+123** | **+4.57** |
| | Unimod. | **−12.9** | **+35.1** | **+183** | **−33.2** | **+38.0** | **+5.66** | **+7.62** | **−13.6** | **+26.8** | **+7.58** |
| FILM | Orig. | 0.120 | 0.956 | 0.015 | 0.413 | 0.270 | 1.53 | 108.7 | 1.09 | 0.179 | **12.59** |
| | Empty | 0.00 | 0.00 | 0.00 | 0.00 | 0.00 | 0.00 | 0.00 | 0.00 | 0.00 | 0.00 |
| | Const. | 0.00 | 0.00 | 0.00 | 0.00 | 0.00 | 0.00 | 0.00 | 0.00 | 0.00 | 0.00 |
| | Shuf. | 0.00 | 0.00 | 0.00 | 0.00 | 0.00 | 0.00 | 0.00 | 0.00 | 0.00 | 0.00 |
| | Cross | 0.00 | 0.00 | 0.00 | 0.00 | 0.00 | 0.00 | 0.00 | 0.00 | 0.00 | 0.00 |
| | Oracle | 0.00 | 0.00 | 0.00 | 0.00 | 0.00 | 0.00 | 0.00 | 0.00 | 0.00 | 0.00 |
| | p=0 | **+21010** | **+670** | **+465** | **+703** | **+99.8** | **+78.3** | **+7.53** | **+294** | **+5069** | **+49.9** |
| | Unimod. | **−11.1** | **+39.9** | **+125** | **−10.4** | **+19.7** | **+30.9** | **+9.42** | **−0.88** | **+49.6** | **+9.88** |
| INFORMER | Orig. | 0.239 | 0.948 | 0.328 | 0.393 | 0.287 | 1.15 | 123.3 | 0.875 | 0.162 | **14.19** |
| | Empty | −0.01 | −0.02 | +0.03 | +0.15 | −0.12 | −0.31 | −0.01 | −0.02 | +0.01 | −0.01 |
| | Const. | +0.05 | +0.04 | +0.01 | +0.12 | +0.16 | +0.03 | 0.00 | +0.02 | +0.02 | 0.00 |
| | Shuf. | +0.10 | 0.00 | 0.00 | +0.12 | +0.05 | +0.09 | 0.00 | +0.01 | +0.03 | 0.00 |
| | Cross | +0.04 | −0.01 | −0.03 | +0.08 | +0.26 | +0.09 | 0.00 | +0.03 | −0.02 | 0.00 |
| | Oracle | +0.13 | −0.02 | +0.02 | +0.17 | **+0.58** | −0.16 | 0.00 | −0.02 | +0.01 | 0.00 |
| | p=0 | **+321** | **+9.64** | **+243** | **−1.83** | **+52.2** | **+8.61** | **+6.21** | **+0.41** | **+44.7** | **+7.54** |
| | Unimod. | **+124** | **+20.3** | **+327** | **−1.77** | **+56.2** | **+23.9** | **+6.98** | **+1.18** | **+53.1** | **+8.38** |
| TRANSFORMER | Orig. | 0.180 | 0.935 | 0.131 | 0.361 | 0.276 | 1.14 | 121.7 | 0.894 | 0.163 | **13.97** |
| | Empty | 0.00 | 0.00 | 0.00 | 0.00 | 0.00 | 0.00 | 0.00 | 0.00 | 0.00 | 0.00 |
| | Const. | 0.00 | 0.00 | 0.00 | 0.00 | 0.00 | 0.00 | 0.00 | 0.00 | 0.00 | 0.00 |
| | Shuf. | 0.00 | 0.00 | 0.00 | 0.00 | 0.00 | 0.00 | 0.00 | 0.00 | 0.00 | 0.00 |
| | Cross | 0.00 | 0.00 | 0.00 | 0.00 | 0.00 | 0.00 | 0.00 | 0.00 | 0.00 | 0.00 |
| | Oracle | 0.00 | 0.00 | 0.00 | 0.00 | 0.00 | 0.00 | 0.00 | 0.00 | 0.00 | 0.00 |
| | p=0 | **+251** | **+14.9** | **+470** | **−20.3** | **+42.7** | **+7.12** | **+4.71** | **+3.05** | **+59.6** | **+5.71** |
| | Unimod. | **+85.0** | **+18.4** | **+501** | **−14.4** | **+39.6** | **+16.9** | **+8.14** | **−1.22** | **+51.0** | **+8.91** |
| iTRANSFORMER | Orig. | 0.094 | 0.996 | 0.010 | 0.302 | 0.260 | 1.30 | 115.5 | 1.07 | 0.195 | **13.31** |
| | Empty | 0.00 | 0.00 | 0.00 | 0.00 | 0.00 | 0.00 | 0.00 | 0.00 | 0.00 | 0.00 |
| | Const. | 0.00 | 0.00 | 0.00 | 0.00 | 0.00 | 0.00 | 0.00 | 0.00 | 0.00 | 0.00 |
| | Shuf. | 0.00 | 0.00 | 0.00 | 0.00 | 0.00 | 0.00 | 0.00 | 0.00 | 0.00 | 0.00 |
| | Cross | 0.00 | 0.00 | 0.00 | 0.00 | 0.00 | 0.00 | 0.00 | 0.00 | 0.00 | 0.00 |
| | Oracle | 0.00 | 0.00 | 0.00 | 0.00 | 0.00 | 0.00 | 0.00 | 0.00 | 0.00 | 0.00 |
| | p=0 | **+23169** | **+60.9** | **+210** | **+251** | **+10.1** | **+52.3** | **+13.3** | **+348** | **+1597** | **+38.4** |
| | Unimod. | **−1.65** | **+21.9** | **+60.5** | **−11.6** | **+6.91** | **+28.5** | **+0.63** | **+9.75** | **+13.0** | **+1.20** |

*Table 10.* **Full MM-TSFLIB results** across eight backbones. **Orig.** row gives the mean test MSE on the original-text baseline; all rows below it report the percentage change in MSE relative to the **Orig.** row of that same backbone, within each domain. **Bold** marks $|\Delta| \geq 0.5\%$; grey marks $|\Delta| < 0.05\%$.

| Backbone | Cond. | Agri | Clim | Econ | Ener | Envi | Heal | Secu | SocG | Traf | Avg. |
|---|---|---|---|---|---|---|---|---|---|---|---|
| | Orig. | 0.105 | 1.19 | 0.072 | 0.364 | 0.567 | 1.90 | 122.1 | 1.06 | 0.213 | **14.17** |
| | Empty | +3.49 | +0.06 | +0.63 | −3.20 | −0.08 | −0.39 | −0.29 | −1.12 | −2.60 | −0.31 |
| | Const. | +1.79 | −0.30 | +9.07 | −0.01 | −1.31 | −1.29 | +0.62 | −1.69 | −0.80 | +0.56 |
| AUTOFORMER | Shuf. | −2.08 | +0.97 | −0.80 | +0.46 | −0.19 | +1.68 | −0.26 | +2.33 | −1.85 | −0.20 |
| | Cross | −2.90 | +1.46 | −12.8 | +2.76 | −0.46 | +0.88 | −0.22 | +1.22 | −0.40 | −0.18 |
| | Oracle | +5.76 | −0.02 | −0.41 | −0.41 | −0.12 | +0.85 | −2.45 | −0.88 | −2.46 | −2.34 |
| | p=0 | +46.8 | +7.64 | +70.2 | −4.26 | +3.25 | +5.50 | −0.17 | −2.73 | −0.57 | +0.05 |
| | Unimod. | +0.49 | +8.94 | +5.71 | −1.33 | +5.84 | +8.66 | +3.68 | −1.01 | +3.99 | +3.76 |
| | Orig. | 0.309 | 1.12 | 0.729 | 0.331 | 0.544 | 1.31 | 126.9 | 0.884 | 0.218 | **14.70** |
| | Empty | −2.92 | +0.51 | −4.03 | +0.41 | +2.28 | +0.13 | +0.02 | −0.92 | −0.61 | 0.00 |
| | Const. | −2.95 | +0.51 | −4.08 | +0.41 | +0.58 | +0.13 | +0.02 | −0.92 | −0.62 | −0.01 |
| CROSSFORMER | Shuf. | +1.30 | −0.14 | −0.98 | 0.00 | +1.09 | 0.00 | 0.00 | −0.02 | 0.00 | 0.00 |
| | Cross | +1.32 | −0.97 | −0.94 | +0.79 | +0.43 | +0.26 | +0.06 | +0.02 | −0.26 | +0.06 |
| | Oracle | +0.95 | +0.04 | −3.94 | +0.29 | +0.80 | +1.65 | +0.21 | −1.04 | −2.50 | +0.19 |
| | p=0 | +25.4 | +4.31 | +31.5 | +2.62 | +8.66 | +4.78 | +0.70 | +2.24 | +6.98 | +1.06 |
| | Unimod. | +12.1 | +4.04 | +20.4 | +1.04 | +8.08 | +4.04 | +0.77 | +1.83 | +8.26 | +1.02 |
| | Orig. | 0.155 | 1.23 | 0.072 | 0.362 | 0.536 | 1.70 | 109.3 | 1.09 | 0.303 | **12.75** |
| | Empty | −0.04 | 0.00 | +0.05 | 0.00 | +0.96 | 0.00 | 0.00 | +0.01 | −0.01 | +0.01 |
| | Const. | −0.05 | 0.00 | −0.51 | 0.00 | +0.96 | 0.00 | 0.00 | +0.01 | −0.03 | 0.00 |
| DLINEAR | Shuf. | 0.00 | +0.01 | −0.48 | +0.21 | +0.95 | 0.00 | 0.00 | +0.01 | −0.02 | +0.01 |
| | Cross | −0.04 | 0.00 | +0.04 | +0.22 | +0.96 | +0.01 | 0.00 | 0.00 | −0.01 | +0.01 |
| | Oracle | −0.03 | +0.01 | +0.02 | +0.22 | +0.93 | +0.15 | 0.00 | +0.01 | −0.05 | +0.01 |
| | p=0 | +745 | +20.1 | +6.94 | +31.0 | −5.54 | +9.58 | +0.65 | −1.85 | +110 | +2.33 |
| | Unimod. | +38.1 | +13.1 | +95.0 | +0.96 | +9.07 | +4.08 | +2.20 | +6.94 | +30.0 | +2.60 |
| | Orig. | 0.093 | 1.15 | 0.045 | 0.267 | 0.493 | 1.41 | 115.7 | 0.946 | 0.180 | **13.36** |
| | Empty | +2.23 | −0.83 | +10.8 | +2.20 | +0.26 | −0.43 | +0.64 | +0.02 | +0.19 | +0.61 |
| | Const. | +2.21 | −0.83 | +10.9 | +2.19 | +0.25 | −0.44 | +0.65 | +0.03 | +0.19 | +0.62 |
| FEDFORMER | Shuf. | −1.31 | +0.03 | +8.09 | 0.00 | +0.14 | +0.01 | −0.01 | +1.11 | +0.01 | 0.00 |
| | Cross | −3.35 | −0.28 | +0.15 | +2.92 | −0.06 | +0.11 | −0.12 | +0.53 | −0.32 | −0.11 |
| | Oracle | +1.27 | −0.07 | −1.41 | −0.50 | +0.02 | −0.36 | +0.86 | −0.59 | −0.65 | +0.82 |
| | p=0 | +39.4 | +0.85 | +136 | +5.87 | +3.73 | +0.43 | −0.15 | −1.36 | +3.97 | −0.03 |
| | Unimod. | +3.16 | +4.65 | +34.1 | +1.83 | +7.11 | +3.23 | +0.13 | 0.00 | +4.29 | +0.27 |
| | Orig. | 0.106 | 1.24 | 0.034 | 0.362 | 0.506 | 1.80 | 116.6 | 1.05 | 0.238 | **13.55** |
| | Empty | +0.32 | −0.03 | −0.05 | +0.44 | −0.05 | +0.01 | +0.01 | +0.34 | −0.03 | +0.01 |
| | Const. | +0.35 | −0.03 | +0.02 | 0.00 | −0.05 | +0.02 | +0.01 | +0.35 | 0.00 | +0.01 |
| FILM | Shuf. | +0.02 | −0.01 | −0.05 | +0.01 | +0.02 | +0.01 | +0.01 | +0.35 | +0.02 | +0.01 |
| | Cross | +0.31 | −0.04 | −0.02 | +0.33 | +0.01 | 0.00 | 0.00 | +0.08 | −0.01 | +0.01 |
| | Oracle | +0.01 | −0.02 | +0.01 | −0.01 | −0.06 | +0.02 | +0.01 | +0.35 | +0.01 | +0.01 |
| | p=0 | +1109 | +21.1 | +0.87 | +43.9 | +9.24 | +6.69 | +0.60 | −0.27 | +165 | +2.34 |
| | Unimod. | +1.24 | +7.75 | −6.83 | −2.58 | +6.42 | +5.70 | +3.34 | +2.60 | +5.31 | +3.40 |
| | Orig. | 0.446 | 1.12 | 0.987 | 0.398 | 0.479 | 1.40 | 129.7 | 0.840 | 0.189 | **15.06** |
| | Empty | −0.53 | +0.11 | +7.70 | +0.43 | −1.38 | +1.88 | −0.04 | +0.33 | +1.52 | +0.11 |
| | Const. | −0.39 | −0.19 | +7.99 | +0.33 | −0.25 | +0.44 | −0.03 | +0.11 | +1.89 | +0.09 |
| INFORMER | Shuf. | −2.51 | 0.00 | +3.93 | −0.19 | −0.75 | +0.16 | +0.10 | +0.72 | +0.30 | +0.12 |
| | Cross | −5.28 | −0.94 | +3.62 | −0.83 | −0.86 | −1.72 | +0.09 | +0.46 | −0.14 | +0.06 |
| | Oracle | +2.06 | +1.10 | +8.07 | +0.73 | −0.28 | +0.22 | +0.07 | +0.42 | +1.84 | +0.15 |
| | p=0 | +13.3 | +0.11 | +17.4 | −1.77 | +2.00 | +4.34 | +1.13 | +1.11 | +7.28 | +1.32 |
| | Unimod. | +12.8 | +3.04 | +22.8 | +1.18 | +3.22 | +2.75 | +1.10 | +0.16 | +6.57 | +1.33 |
| | Orig. | 0.287 | 1.09 | 0.452 | 0.330 | 0.459 | 1.29 | 130.8 | 0.851 | 0.175 | **15.08** |
| | Empty | −5.03 | −1.41 | +4.70 | +4.18 | +0.29 | +3.18 | −0.06 | −0.40 | −1.05 | −0.03 |
| | Const. | −5.05 | −1.40 | +4.78 | +4.17 | +0.29 | +3.19 | −0.05 | −0.42 | −1.00 | −0.01 |
| TRANSFORMER | Shuf. | −1.68 | −1.05 | +1.15 | −0.01 | −0.16 | +0.01 | +0.03 | +1.48 | +0.03 | +0.03 |
| | Cross | −0.10 | −1.95 | +2.44 | −1.29 | −0.29 | +0.77 | +0.02 | +1.10 | −0.75 | +0.02 |
| | Oracle | −0.27 | −1.36 | +0.12 | +0.48 | +0.18 | +1.25 | +0.07 | −0.38 | +0.03 | +0.07 |
| | p=0 | +14.0 | +0.80 | +32.6 | −4.60 | +4.05 | +6.68 | +1.23 | +4.19 | +5.22 | +1.43 |
| | Unimod. | +13.1 | +2.87 | +18.2 | −3.84 | +1.76 | +6.23 | +1.10 | +3.57 | +6.57 | +1.25 |
| | Orig. | 0.091 | 1.14 | 0.018 | 0.274 | 0.417 | 1.63 | 117.1 | 1.19 | 0.210 | **13.56** |
| | Empty | −0.30 | −0.29 | −0.59 | +0.16 | −0.37 | −1.06 | +0.06 | −3.30 | +0.62 | +0.01 |
| | Const. | −0.25 | −0.29 | −0.69 | −0.18 | −0.36 | −0.55 | +0.06 | −2.95 | +0.14 | +0.02 |
| iTRANSFORMER | Shuf. | −0.17 | −1.27 | −0.49 | −0.07 | −0.36 | +0.01 | −0.03 | −1.36 | −0.02 | −0.05 |
| | Cross | +0.46 | +0.58 | +0.25 | −0.85 | −0.36 | −0.35 | −0.07 | −2.12 | −0.16 | −0.09 |
| | Oracle | +0.25 | −1.08 | +0.35 | −1.09 | +0.10 | −0.39 | −0.01 | +3.72 | +0.32 | +0.01 |
| | p=0 | +539 | +3.39 | +37.5 | +29.7 | +1.77 | +10.6 | +2.90 | +32.8 | +27.1 | +3.80 |
| | Unimod. | +1.16 | +5.67 | −9.96 | +1.62 | +2.49 | +5.95 | +1.33 | +3.17 | +2.72 | +1.45 |

*Table 11.* **Full paired bootstrap CIs** for all (model, condition) pairs. $n$ is the number of matched (condition, original) cell pairs; $\Delta\%$ is the ratio-of-means change; the 95% CI is the $[2.5\%, 97.5\%]$ percentile of the bootstrap distribution; $p$ is the two-sided bootstrap $p$-value. The first block under each model gives the four cells of the factorial in Table 3 that vary text with **p** intact. The second block gives the cells with **p** zeroed. **Bold** marks $p < 0.001$.

| Model | Condition | $n$ | $\Delta\%$ | 95% CI | $p$ |
|---|---|---|---|---|---|
| | Empty (col) | 108 | +0.024 | $[+0.005, +0.040]$ | 0.019 |
| | Const (col) | 108 | +0.030 | $[-0.027, +0.103]$ | 0.296 |
| | Shuffled | 108 | +0.004 | $[-0.001, +0.009]$ | 0.110 |
| | Cross-domain | 108 | −0.006 | $[-0.031, +0.022]$ | 0.657 |
| AURORA | Oracle | 108 | +0.024 | $[-0.026, +0.085]$ | 0.347 |
| | Empty (col=0) | 108 | +0.024 | $[+0.005, +0.040]$ | 0.019 |
| | Const (col=0) | 108 | +0.030 | $[-0.027, +0.103]$ | 0.296 |
| | Col zeroed | 108 | 0.000 | $[\ 0.000,\ 0.000]$ | 1.000 |
| | Unimodal | 108 | 0.000 | $[\ 0.000,\ 0.000]$ | 1.000 |
| | Empty (col) | 864 | +0.048 | $[-0.251, +0.349]$ | 0.750 |
| | Const (col) | 864 | +0.157 | $[-0.148, +0.467]$ | 0.322 |
| | Shuffled | 864 | −0.010 | $[-0.083, +0.050]$ | 0.847 |
| | Cross-domain | 864 | −0.027 | $[-0.202, +0.144]$ | 0.731 |
| MM-TSFLIB | Oracle | 864 | −0.142 | $[-0.508, +0.196]$ | 0.449 |
| | **Empty (col=0)** | 864 | **+1.624** | $[+1.249, +2.021]$ | **<0.001** |
| | **Const (col=0)** | 864 | **+1.632** | $[+1.254, +2.033]$ | **<0.001** |
| | **Col zeroed** | 864 | **+1.517** | $[+1.212, +1.852]$ | **<0.001** |
| | **Unimodal** | 864 | **+1.868** | $[+1.495, +2.297]$ | **<0.001** |
| | Empty (col) | 864 | −0.001 | $[-0.002, +0.001]$ | 0.321 |
| | Const (col) | 864 | −0.000 | $[-0.001, +0.000]$ | 0.511 |
| | Shuffled | 864 | −0.000 | $[-0.001, +0.001]$ | 0.473 |
| | Cross-domain | 864 | +0.000 | $[-0.001, +0.002]$ | 0.822 |
| TATS | Oracle | 864 | +0.000 | $[-0.001, +0.002]$ | 0.991 |
| | **Empty (col=0)** | 864 | **+20.165** | $[+16.768, +24.720]$ | **<0.001** |
| | **Const (col=0)** | 864 | **+20.164** | $[+16.767, +24.720]$ | **<0.001** |
| | **Col zeroed** | 864 | **+20.164** | $[+16.767, +24.719]$ | **<0.001** |
| | **Unimodal** | 864 | **+7.552** | $[+6.425, +8.760]$ | **<0.001** |

