# OpenReview forum: "Semantics or Structure? Auditing Text Sensitivity in Multimodal Time-Series Forecasting"
_ICML.cc/2026/Workshop/FMSD — FMSD @ ICML 2026 Poster_

### Official Review · Reviewer_xTd8 · 2026-05-19
**Interesting findings about text sensitivity in multi-modal forecasting models**

**Rating:** 8
**Confidence:** 3

**Review:**

Summary

The paper evaluates three multimodal approaches for time-series forecasting and investigates whether their reported gains actually originate from the textual modality. The authors apply several controlled text perturbation strategies and show experimentally that these multimodal approaches exhibit very limited sensitivity to the semantic content of text. For each architecture, the paper further analyzes the source of the observed improvements over unimodal baselines and identifies alternative explanations, including architectural and implementation factors.

Strengths

Clear and important research question: do multimodal forecasting gains actually come from textual semantics?
The paper is easy to read and follows a logical structure.
Claims are supported by rigorous experimentation with multiple perturbation types.
Strong mechanistic analysis beyond simple performance comparisons.
Interesting and potentially important findings regarding the actual behavior of multimodal time-series forecasting models.
The analysis extends across multiple architectures rather than a single model, increasing the relevance of the conclusions.

Areas for Improvement

The paper could cite concurrent work on multimodal shortcut learning in other domains (for example vision-language models exploiting dataset biases) to situate the findings within a broader pattern.

The oracle experiment may not provide a strong test of semantic sensitivity since frozen text encoders often struggle with numerical reasoning.

Since all evaluated approaches rely on frozen text encoders, it remains unclear whether the observed limitations arise from the benchmark itself or from architectural choices.

Detailed Comments

The paper could better position itself with prior work on shortcut learning and modality collapse in multimodal learning. Similar observations have been made in vision-language systems where models rely on unintended signals or dataset biases rather than the intended modality.

The paper argues that low semantic diversity in Time-MMD may contribute to the observed behavior. However, this remains correlational. An experiment artificially increasing semantic diversity through text augmentation would strengthen this argument.


Justification of Score

The paper addresses an important and underexplored question in multimodal time-series forecasting: whether reported multimodal gains genuinely arise from semantic use of textual information. The work goes beyond standard performance comparisons by introducing controlled perturbation experiments and conducting mechanistic analyses to explain the observed behavior of multiple architectures. The experimental design is rigorous, covering multiple perturbation strategies and several multimodal approaches, which makes the conclusions convincing. The findings are also interesting and potentially impactful, as they challenge assumptions commonly made in recent multimodal forecasting literature and may influence how future benchmarks and evaluation protocols are designed.

The paper is also clearly written and easy to follow. Claims are generally supported by empirical evidence rather than speculation, and the additional investigation into implementation details and architectural behavior strengthens the overall contribution.

Overall, the paper presents a strong and well-executed study with meaningful findings and clear practical implications, making it a clear accept.

---

### Official Review · Reviewer_corR · 2026-05-20

**Rating:** 7
**Confidence:** 4

**Review:**

# Summary
This paper investigates whether the performance gains reported on the multimodal time-series forecasting benchmark Time-MMD are actually driven by semantic information from text inputs. The authors conduct a systematic perturbation study by replacing the original text with empty, shuffled, cross-domain, and oracle texts, and show that these substitutions lead to only negligible performance changes. The study further reveals that a substantial portion of the multimodal gain comes from auxiliary numerical information such as prior_history_avg, suggesting that the text modality in Time-MMD may not be sufficiently informative in practice.

# Strengths
- The perturbation-based evaluation protocol is clear and convincing, and effectively isolates the source of multimodal gains.
- The paper goes beyond simple benchmarking and carefully analyzes the role of prior_history_avg and architectural factors contributing to the observed improvements.
- The experiments are reasonably extensive and the reproducibility details are well documented.
- The work raises an important and meaningful question regarding the validity of current multimodal TSF benchmarks and evaluation practices.

# Weaknesses
- The analysis is limited to the Time-MMD benchmark, so it remains unclear whether the findings generalize to other datasets or newer multimodal forecasting architectures.
- The oracle text and numeracy-related experiments are somewhat limited; stronger directional oracle settings could further strengthen the conclusions.
- Some parts of the fusion equation and implementation descriptions are slightly confusing and could be clarified.

# Overall Assessment
This is an interesting and meaningful analysis paper that critically revisits the reported gains in multimodal forecasting on Time-MMD. The systematic investigation into whether text semantics are actually utilized is particularly valuable and provides important implications for benchmark design and multimodal evaluation practices. Although the scope is somewhat limited, I believe the paper is sufficiently valuable for a workshop setting.

---

### Official Review · Reviewer_y5T7 · 2026-05-22
**Review of paper 187**

**Rating:** 6
**Confidence:** 5

**Review:**

# Summary
This paper presents a rigorous audit of multimodal time-series forecasting models (including AURORA, MM-TSFLIB, and TATS) on the Time-MMD benchmark. Through controlled text perturbations—replacing real text with empty strings, constant placeholders, within-domain shuffled text, cross-domain text, or numerical future values (oracle)—the authors demonstrate that none of the models exhibit sensitivity to semantic content, as downstream Mean Squared Error (MSE) changes by less than 0.5% in all conditions. Mechanistic localization reveals that the reported "multimodal lift" in TATS and MM-TSFLIB is almost entirely driven by a co-shipped numerical rolling average column (`prior_history_avg`) blended directly into the fusion residual rather than the text encoder. For AURORA, which does not consume this numerical feature, gradient and attention entropy probes show that the text pathway, while optimized during training, remains content-blind in the forward pass due to the extremely low semantic diversity (SDI) of the benchmark's text. The paper concludes that existing multimodal forecasting benchmarks do not verify genuine text utilization and proposes a diagnostic perturbation harness for future research.

## Pros
- **Rigorous and Comprehensive Perturbation Protocol:** The introduction of a five-condition text perturbation protocol (Empty, Constant, Shuffled, Cross-domain, and Oracle) is a highly clever and systematic methodology to isolate specific facets of text presence, token semantics, temporal alignment, and topical relevance. This provides the research community with an invaluable, reusable diagnostic toolkit to expose "Clever Hans" phenomena and ensure genuine multimodal integration in future benchmarks.
- **Elegant and Insightful Mechanistic Localization:** The authors go far beyond simply documenting content insensitivity; they mathematically and empirically trace the exact source of the artificial "multimodal lift." Through precise algebraic analysis of fusion interfaces in TATS and MM-TSFLIB (Equations 1 & 2), the paper demonstrates that the reported performance gains are almost entirely driven by the co-shipped numeric prior column `prior_history_avg` (`p`) sharing the same residual fusion pipeline. This rigorous debugging provides a definitive and highly valuable explanation for the field's misleading benchmarks.

## Cons
- **Exclusively Restrained to Time-MMD Benchmark:** The entire audit is conducted solely on the Time-MMD benchmark. While Time-MMD is currently the primary standard in this subfield, its exceptionally low Semantic Diversity Index (SDI) means the observed content-blindness may be a symptom of this specific benchmark's poor text quality rather than an inherent architectural limitation of multimodal time-series models. Auditing alternative tasks with high-diversity text (e.g., financial forecasting with diverse news corpora) is necessary to evaluate the models' true semantic capacities.
- **Lack of Concrete Architectural Solutions:** The paper is purely diagnostic and does not propose or validate a concrete architectural fix or model modification that successfully leverages textual semantics. While it offers high-level recommendations (such as training encoders end-to-end or using higher-diversity datasets), it leaves the actual implementation and empirical verification of these solutions entirely to future work.